# Are Essential Women’s Healthcare Services Fully Covered? A Comparative Analysis of Policy Documents in Shanghai and New York City from 1978–2017

**DOI:** 10.3390/ijerph18084261

**Published:** 2021-04-17

**Authors:** Qingyu Zhou, Qinwen Yu, Xin Wang, Peiwu Shi, Qunhong Shen, Zhaoyang Zhang, Zheng Chen, Chuan Pu, Lingzhong Xu, Zhi Hu, Anning Ma, Zhaohui Gong, Tianqiang Xu, Panshi Wang, Hua Wang, Chao Hao, Li Li, Xiang Gao, Chengyue Li, Mo Hao

**Affiliations:** 1Research Institute of Health Development Strategies, Fudan University, Shanghai 200032, China; qingyu.zhou@hotmail.com (Q.Z.); 19211020038@fudan.edu.cn (Q.Y.); 18211020095@fudan.edu.cn (X.W.); yiran_eric@126.com (L.L.); gaoxiang_87@163.com (X.G.); 2Collaborative Innovation Center of Social Risks Governance in Health, Fudan University, Shanghai 200032, China; pwshi@163.com (P.S.); shenqunhong108@163.com (Q.S.); zhangzhy@nhc.gov.cn (Z.Z.); chenzhengjd@163.com (Z.C.); puchuan68@sina.com (C.P.); lzxu@sdu.edu.cn (L.X.); aywghz@126.com (Z.H.); yxyman@126.com (A.M.); zhgong_zg@163.com (Z.G.); xtq1960@icloud.com (T.X.); wangpanshi03@163.com (P.W.); jswstwh@163.com (H.W.); 18906113216@189.cn (C.H.); 3Department of Health Policy and Management, School of Public Health, Fudan University, Shanghai 200032, China; 4Zhejiang Academy of Medical Sciences, Hangzhou 310012, China; 5School of Public Policy and Management, Tsinghua University, Beijing 100084, China; 6Project Supervision Center of National Health Commission of the People’s Republic of China, Beijing 100044, China; 7Department of Grassroots Public Health Management Group, Public Health Management Branch of Chinese Preventive Medicine Association, Shanghai 201800, China; 8School of Public Health and Management, Chongqing Medical University, Chongqing 400016, China; 9School of Public Health, Shandong University, Jinan 250012, China; 10School of Health Service Management, Anhui Medical University, Hefei 230032, China; 11School of Management, Weifang Medical University, Weifang 261053, China; 12Committee on Medicine and Health of Central Committee of China Zhi Gong Party, Beijing 100011, China; 13Institute of Inspection and Supervision, Shanghai Municipal Health Commission, Shanghai 200031, China; 14Shanghai Municipal Health Commission, Shanghai 200031, China; 15Jiangsu Preventive Medicine Association, Nanjing 210009, China; 16Changzhou Center for Disease Control and Prevention, Changzhou 213003, China

**Keywords:** women’s healthcare, service coverage, service assessment, Shanghai, New York City, maternal mortality ratio

## Abstract

This study aimed to analyze the changes in the 10 major categories of women’s healthcare services (WHSs) in Shanghai (SH) and New York City (NYC) from 1978 to 2017, and examine the relationship between these changes and maternal mortality ratio (MMR). Content analysis of available public policy documents concerning women’s health was conducted. Two indicators were designed to represent the delivery of WHSs: The essential women’s healthcare service coverage rate (ESCR) and the assessable essential healthcare service coverage rate (AESCR). Spearman correlation was used to analyze the relationship between the two indicators and MMR. In SH, the ESCR increased from 10% to 90%, AESCR increased from 0% to 90%, and MMR decreased from 24.0/100,000 to 1.01/100,000. In NYC, the ESCR increased from 0% to 80%, the AESCR increased from 0% to 60%, and the MMR decreased from 24.7/100,000 to 21.4/100,000. The MMR significantly decreased as both indicators increased (*p* < 0.01). Major advances have been made in women’s healthcare in both cities, with SH having a better improvement effect. A common shortcoming for both was the lack of menopausal health service provision. The promotion of women’s health still needs to receive continuous attention from governments of SH and NYC. The experiences of the two cities showed that placing WHSs among policy priorities is effective in improving service status.

## 1. Introduction

Women’s health is fundamental to the prosperity of a country. Maintaining women’s health is of great importance for the development of society. In 1978, “Maternal and child health care, including family planning” was identified as one of the eight essential services of primary healthcare through The Alma-Ata Declaration issued by the World Health Organization (WHO) and the United Nations International Children’s Emergency Fund (UNICEF) [1]. In 2000, “Improving maternal health” was one of the eight Millennium Development Goals in the United Nations Millennium Declaration [2]. “Reducing the global maternal mortality ratio (MMR) to less than 70 per 100,000 live births by 2030” ranked the first among the nine indicators in Goal 3 of the 17 Sustainable Development Goals (SDGs) in the 2030 Agenda for Sustainable Development, adopted by all United Nations member states in 2015 [3]. This is a global target that both developed and developing countries are striving to achieve. However, as of 2015, 92 of the 181 countries (50.8%) whose MMR was available for WHO World Health Statistics still did not meet the target [4].

A growing body of evidence has shown that providing women’s healthcare services (WHSs) plays an important role in improving women’s health [5]. High availability and accessibility of WHSs could reduce MMR and the occurrence of postpartum complications [6], as well as promote disease prevention and improve women’s health conditions [7]. For instance, Sweden was the first country to provide home delivery services by professional midwives. This initiative enabled Sweden to reduce its MMR to 228 per 100,000 live births in 1900, which was the lowest in Europe at the time [8]. Sweden continued to reduce its MMR to 4 per 100,000 live births in 2015 [4]. In China, hospital delivery rate across the country increased by 2.1% annually from 2001 to 2014, and MMR decreased by 6.25% annually over the same period [9]. However, the provision of WHSs varies in different countries and regions, leading to regional disparities in women’s health. Inadequate provision of prenatal healthcare services can increase the risk of urinary tract infections and gestational hypertension during pregnancy [10]. In Peru, 87% of the pregnant women had access to prenatal care services at least four times during their pregnancies, but the coverage rate of prenatal care services was only 12% in Ethiopia in 2005. As a result, at the time, MMR in Ethiopia was 720 per 100,000 live births, higher than that in Peru (240 per 100,000 live births) [11]. The authors believe that an investigation of different coverages and practices of WHSs across regions and cities will help demonstrate best practices and shortcomings, and contribute to the improvement of WHSs provisions.

There have been many studies involving the analysis of WHSs utilization [12,13], factors that influences WHSs delivery levels [14,15], and trends of WHSs development [16,17]. Most of these studies have focused on indicators such as implementation [18], management technology and standard [19], coverage rate [20], service equity [21,22], service efficiency [23], and service quality [24]. Some interventional studies have analyzed the improvement on the utilization of WHSs and women’s health after a policy is implemented [25,26]. In addition, a few studies have quantitatively analyzed the relationship between the delivery of WHSs and maternal health outcomes, individual factors such as advanced age, ethnic minority, low education level, and lack of care could significantly affect the risk of preterm birth and stillbirth [27,28]. Receiving adequate prenatal care services could increase the possibility of newborn maintaining a healthy weight [29]. Increased utilization of health services such as prenatal screening and hospital delivery could lead to a significant reduction in MMR [30]. However, most of these studies used annual or short-term data (predominantly less than 20 years) from population surveys, hospitals/women’s healthcare institutions, or statistical yearbooks to analyze the trends and influencing factors of certain categories of WHSs or examine the impact of certain interventions on women’s health. However, to our knowledge, little evidence exists that quantitatively analyzes the extent to which all essential WHSs have been covered and their association with women’s health.

Shanghai (SH) in China and New York City (NYC) in the USA are both important megacities. According to the GaWC 2018 compiled by Globalization and World Cities Study Group and Network, NYC is an Alpha++ global city and SH is an Alpha+ global city [31]. These two cities have developed a relatively complete WHSs systems. SH has established a specialized WHSs system according to the unified national requirements. In addition to community health service centers and hospitals, there are also centers for women and children’s health acting as specialized public health institutions [32]. In NYC, WHSs system is nested within the healthcare system. General practitioners, communities, hospitals and other providers working together in a network structure [33]. In 2017, MMR in SH was 3.0 per 100,000 live births, which was the lowest in China. And the MMR in NYC was 21.5 per 100,000, which was well bellowed the SDGs’ target but higher than the national average [34], that was attributed to racial disparity by some studies [35]. While SH also has disparity between natives and migrants as a multi-ethnic settlement in China [36]. It is worth exploring whether WHSs delivery has an impact on MMRs in the two cities. Some studies on WHSs have selected SH and NYC as the research setting. For example, Perloff analyzed the utilization and influencing factors of prenatal care services in NYC. The study showed that residence in a distressed urban neighborhood lead about 15% of pregnant women to enter prenatal care late in 1990 [37]. Joyce found that the Prenatal Care Assistance Program in New York State could improve birth outcomes and reduce the low birthweight rate by 1.3% in 1991 [38]. Liu et al. found disparity between natives and migrants in SH, and analyzed the barriers in maternal health service utilization in 2009 [39]. Li et al. analyzed and compared the perspective of regulatory policies on MMRs control in SH and NYC, concluding that the more comprehensive and effective regulatory policies had a more substantial impact on MMR control from 2006 to 2017 [40]. We have seen few studies on the impact of WHSs delivery on MMRs in the two cities.

The purpose of this study was to quantify the coverage of essential WHSs and the trends over a 40-year span beginning 1978 (the year that “Primary Health Care for All” was initiated by the WHO) through a content analysis of policies of WHSs delivery in SH and NYC, and to analyze their impacts on women’s health outcomes. The results will help to identify what categories of services are currently insufficient and promote the improvement of the essential WHSs to achieve full coverage of women’s health needs, which could be of great benefit to women’s health status in the two cities. The lessons of the two cities could provide references for countries around the world, especially developing countries, to achieve the SDGs’ target.

## 2. Materials and Methods

### 2.1. Ethical Approval

In this study, all the data was from public policy documents and official reports. It is an evaluation on WHSs delivery without interventionary studies involving humans. Therefore, the study was exempt from ethical approval.

### 2.2. Study Design and Setting

A comparative policy analysis of documents was conducted to understand the delivery of WHSs and its impact on women’s health outcomes between 1978 and 2017. We selected SH and NYC as the research setting. The two cities have similar economic and social statuses in their respective counties. SH and NYC are the economic and financial centers of China (the world’s largest developing country) and USA (the world’s largest developed country), with a per capita GDP of 18,761 USD and 85,891 USD in 2017, being the highest city in the respective country. They also have the largest population in built-up areas in their respective countries with similar life expectancies (SH, 82.5 years; NYC, 81.2 years), sex ratios (SH, 98.3; NYC, 91.2) and live birth rates (SH, 8.1‰; NYC, 13.6‰) in 2017 [41,42]. In addition, both SH and NYC have taken strengthening the provision and quality of WHSs as an important issue the health sector agenda [43,44], and the two cities require full coverage for insurance on WHSs [45,46]. Moreover, both SH and NYC have consistently performed well in terms of WHSs delivery. For example, the percentage of women under systematic maternal management exceeded 96.2% in SH in 2017 [47]. Simultaneously, the percentage of women with a preventive medical visit in the past year in NYC has increased to 79.1% [48].

The focus of this study is to examine the evolution of women’s health services policies in SH and NYC and its impact on women’s health outcomes in these two cities. In the context of achieving SDGs’ target, examining their policy developments in WHSs may help provide a reference worldwide.

MMR was selected as an indicator of women’s health in this study, as it is the most commonly used and representative indicator for the measurement of women’s health, and it is one of the three indicators that reflects the health of the population recognized internationally. Compared with other outcome indicators, MMR has been continuously monitored in SH and NYC, which is beneficial to this study.

### 2.3. Definition of Essential Women’s Healthcare Service

Based on the guidelines and agendas for women’s health issued by the WHO and countries such as China, the USA, and the UK [49,50,51,52,53,54], and consulting experts, we selected 10 main categories of WHSs. These 10 services included unintended pregnancy intervention, safe abortion, pre-marital healthcare, family planning, infertility interventions, high-risk pregnancy screening, prenatal care, intrapartum care, postpartum care, and menopausal care. These 10 services covered the complete life-cycle of women from adolescence, childbearing age, prenatal and postnatal period to menopause (Table 1).

### 2.4. Measures

Two indicators we adopted in this study to assess the WHSs delivery were the essential women’s healthcare service coverage rate (ESCR) and the assessable essential healthcare service coverage rate (AESCR).

ESCR evaluated the number of categories of WHSs provided in the city, which represented the scope of service coverage. The value of this indicator was calculated by dividing the number of services provided by the number of services that should be provided [55]. The study assumed that a higher number of services provided was indicative that women’s health needs were better met.

AESCR evaluated whether the WHSs could be assessed with corresponding assessment indicators or assessment criteria. The value of this indicator was calculated by dividing the number of WHSs that have been set using assessment criteria by the number of categories of services that should be provided [56]. The study assumed that setting assessment criteria means specifying requirements such as the quantity, frequency, or level of service delivery. It could better ensure the quality of the services.

### 2.5. Data Collection

#### 2.5.1. Policy Documents Collection

Healthcare for women is a part of the public health services. The public has the right to obtain information about health services and programs provided by the government or public organizations (such as maternal and child healthcare centers in China) through an open source. This study used content analysis to code the relevant contents mentioned in policy documents about women’s healthcare to examine the delivery and service assessment of WHSs in the two cities. The documents included for this analysis were policy documents, from 1978 to 2017, related to women’s health in SH and NYC.

The types of policy documents used for analysis included bills, laws, regulations, strategies, norms, rules, plans, budgets, guidelines, and standards. All documents were collected from the official websites of the legislature (Shanghai Municipal People’s Congress, http://www.spcsc.sh.cn/; New York City Council, https://council.nyc.gov/, etc.), governmental organizations (Shanghai Municipal Health Commission, https://wsjkw.sh.gov.cn/; NYC Department of Health and Mental Hygiene, https://www1.nyc.gov/site/doh/index.page, etc.), maternal and child healthcare agencies (Shanghai Center for Women and Children’s Health, http://www.shmchc.cn/, etc.) and legal databases (Law Library, law-lib.com; Thomson Reuters Westlaw, https://govt.westlaw.com) under the principles of “as much as possible” and “publicly released”. All the documents were filtered according to the exclusion criteria which were duplicates, non-binding, without administrative validity such as news and without specific release date. In total, 417 documents about SH were collected, and 301 documents about NYC were collected.

#### 2.5.2. Assessing Information Collection

The members of the research team, after training, coded each document individually. The data included basic information of the documents and the content that were relevant to women’s healthcare. The basic information included the name of the document, the type of document, the year of publication, and the department or institution that issued it. The other part included information such as “service,” “intervention,” “measures,” “initiatives,” “indicators,” “assessment,” and “evaluation,” was used to determine the categories of WHSs and how the WHSs would be assessed.

We calculated the number of services that had been provided according to the documents, and the number of WHSs that had been set using assessment standards. This was further coded according to the documents which mentioned “assessment indicators,” “assessment requirements,” and “evaluation”.

We analyzed the credibility of the data collection using the test-retest reliability method with the intra-group correlation coefficient (ICC). After retesting by two experienced researchers on the number of categories of services and the number of service types with assessment indicators, the ICCs were 0.993 and 0.932, respectively. The values were found to be greater than 0.750, indicating high credibility.

#### 2.5.3. Collection of MMRs Data

The MMRs of the population of households registered in SH from 1978 to 2017 were collected from the Shanghai Statistical Yearbook online [41]. The MMRs of NYC from 1978 to 2017 were collected from the summary of Vital Statistics for NYC [57].

### 2.6. Statistical Analysis

We used Microsoft Excel 2010 (Microsoft, Redmond, WA, USA) to create and process the database. We used SPSS 13.0 (SPSS Inc., Chicago, IL, USA) for statistical analysis.

Vertical analysis was used to study the trends of ESCR, AESCR, and MMR from 1978 to 2017 in the two cities. We adopted the Mann-Whitney U test to compare the difference between the two cities and used Spearman correlation to analyze the influence of ESCR and AESCR on MMR. All *p* values were double-sided, and the confidence interval was 95%. A *p* value of <0.05 was considered significant.

## 3. Results

### 3.1. The Trends of MMRs in SH and NYC

MMR in SH showed a significant declining trend (Figure 1) from 24.0/100,000 to 15.9/100,000 during 1978–1986; it grew from 1987 to 1988, and reached the max value of 33.06/100,000 in the past four decades in 1988. After 1994, it showed a gradual downward trend, from 30.29/100,000 to 1.01/100,000 in 2017.

MMR in NYC fluctuated over the past 40 years (Figure 2). It showed a declining trend during the period 1978–1998, when the MMR ranged from 12.9/100,000 to 24.7/100,000, except for a significant increase to 39.2/100,000 in 1983. We could divide the years into three stages after 1998, 1998–2005, 2005–2012, and 2012–2017. The MMR in 2001 (33.1/10 million), 2008 (30.5/100,000), and 2015 (28.8/100,000) were the highest in each stage.

The MMR was sometimes higher in SH and sometimes higher in NYC between 1978 and 1997. Since 1998, the MMR in SH was significantly lower than that in NYC. Overall, the MMR in SH was lower than that in NYC (*p* <0.01).

### 3.2. The Trends of ESCR in SH and NYC

The trend of ESCR in SH is shown in Figure 1. In the mid-to-late 1970s, China began to carry out family planning to improve fertility quality, at that time, the ESCR was 10%. In 1985, the ESCR increased to 50%, as the WHSs package was launched nationwide by the Ministry of Health [58]. The package covered four categories of essential services including high-risk pregnancy management, prenatal care, intrapartum care and postnatal care. In 1986, the Ministry of Health promoted contraceptive methods throughout the country to reduce the rate of artificial abortion and unintended pregnancy [59]. In 1992, pre-marriage health screening was launched nationwide, and the ESCR reached 70% [60]. Since 1998, SH has provided prevention, treatment, and consultation services of infertility for people of reproductive age [61], and the ESCR has increased to 80%. In 2001, the ESCR increased to 90%, as unintended pregnancy prevention services were provided through methods including decision-making tools for family planning clients and providers, and free distribution of contraceptives.

The trend of the ESCR in NYC is shown in Figure 2. In 1979, a regulation was published to optimize the timing of pregnancies, provide services for prevention of unintended pregnancies, and genetic diagnosis of high-risk pregnant women [62]. It also put forward requirements for prenatal care, intrapartum care, and postpartum care; The ESCR was 60% at that time. The *Preventive Health Amendments of 1992* was published and it required the screening and provision of treatment for infectious diseases and maternal diseases that could cause infertility in women. Following that, the ESCR increased to 70%. In 2012, abortion rate was incorporated into maternal and child health indicators in the *New York State Community Health Indicator Reports*, and the ESCR increased to 80%.

From 1978 to 1986, ESCR was higher in NYC than in SH. After 1986, the ESCR in SH grew rapidly and exceeded that in NYC. In general, SH was better than NYC (*p* = 0.018) in this aspect. On the other hand, public health services for menopausal care have not been provided in either city.

### 3.3. The Trends of AESCR in SH and NYC

The trend of the AESCR in the SH is shown in Figure 1. In 1985, the Ministry of Health clarified the assessment requirements for prenatal care, intrapartum care, and postnatal care (such as antenatal inspection rate and postpartum visit rate) [58], and the AESCR reached 30%. In the mid-1990s, assessment indicators for pre-marriage health screening, family planning, and high-risk pregnancy healthcare were set by the government of China [60,63], and the AESCR increased to 60%. In 1998 and 2001, quantitative assessment indicators such as treatment rate and service coverage rate were set up for three services, including infertility intervention, abortion, and unintended pregnancy prevention in SH, and the AESCR shot up to 90%.

According to the available documents, there were fewer quantitative assessment standards or indicators for WHSs in NYC. In 2008, NYC made explicit requirements for the control of maternal deaths and listed them as priorities. In 2010, a clear requirement was made for the pregnancy rate of adolescents [64]. In 2012, The *New York State Community Health Indicator Reports* set the indicators of the pregnancy rate of women, the proportion of prenatal screening for maternal, and the screening proportion of high-risk pregnancy, which led the AESCR in NYC to reach 60%. Overall, the AESCR in the SH was higher than that in NYC (*p* < 0.01).

### 3.4. The Relationship between ESCR, AESCR, and MMR in SH and NYC

The MMRs in both SH and NYC decreased with the increase of ESCR and AESCR, suggesting that the trends between them might be correlated (Table 2). Correlation analysis showed that the ESCR and AESCR, was negatively correlated with MMR in SH (coefficients = −0.826 and −0.835, *p* < 0.01), and that the ESCR in NYC was negatively correlated with MMR (coefficient = −0.324, *p* = 0.041).

## 4. Discussion

To our knowledge, this study is one of the first to use public policy documents, instead of data from surveys on the utilization of women’s healthcare, to quantitatively analyze of the 10 essential WHSs provided in SH and NYC over the past 40 years, as well as their relationship with MMR to provide evidence for the further improvement of the service in both the cities.

The results demonstrated increasing trends of ESCR and AESCR over the past 40 years, suggesting that service quantity and quality have improved significantly during this period both in SH and NYC. This was consistent with the conclusions of other researchers who analyzed WHSs in SH and NYC [65,66], and found that an increasing number of women could easily access high-quality healthcare services [67]. On one hand, the UN and WHO formulated agendas or proposals such as “Primary Health Care”, “reproductive health care for all age-appropriate people”, and “Millennium Development Goals” calling on countries to ensure adequate women’s healthcare. On the other hand, various countries and cities, including SH and NYC, prioritized women’s health and provided WHSs that covered different life stages of women [29]. For example, in SH, the government has always attached importance to women ’s health and promoted the mission, with a slogan of “the Higher and the Lower”, to improve birth safety and quality and reduce MMR [68]. The municipal government continued to increase investment in women’s health and to promote further education and clinical skills training of women’s healthcare professionals [69]. Furthermore, the municipal government has issued a series of criteria and guidelines to clarify the provision and assessment requirements of the services for high-risk maternal screening, antenatal check-ups, and infertility interventions [70]. The delivery of WHSs in NYC, like other medical services, was mainly provided by general practitioners through primary care centers. Some documents such as *New York State Breastfeeding Mothers’ Bill of Rights* were issued to ensure local residents could access essential WHSs [71].

Both the ESCR and the AESCR are lower in NYC than SH in 2017. These might be due to different policy priorities of the WHSs in the two cities. On one hand, NYC did not have a policy requirement for pre-marital healthcare. The Department of Health of New York State explicitly states that “no pre-marital examination or blood test is required to obtain a marriage license in New York State” [72]. SH was different, as the *Marriage Law of the People’s Republic of China* states that “a marriage shall be prohibited under the circumstance where the man or the women is suffering from any disease which is regarded by medical science as rending a person unfit for marriage” [73]. The accompanying laws and regulations, such as *the Maternal and Child Health Law*, the *Regulations on the Implementation of the Maternal and Child Health Law*, and *the Regulations on Marriage Registration*, ensured access to pre-marital healthcare, especially medical examinations. This led to the difference in the choice of policy priorities between the two cities. Some studies have shown that policy for pre-marital healthcare is closely related to the level of economic and social development [74]. On the other hand, more WHSs could be assessed with corresponding assessment indicators or criteria on coverage in SH. Take postpartum family visits as an example, the *1985 National Quality Standards and Requirements for Maternal Health in Urban and Rural Areas* required that the percentage of women who take postpartum visits (at least three times) needed to reach 80% [58]. Similar standards and requirements were found in several documents since then. By 2017, the percentage has reached 98% in SH [47]. NYC had similar services, such as some kinds of the family support programs, provided a richer range of services free of charge to families in need, including not only healthcare services, but also social supports [75]. The federal government also tried to promote the expansion of these services through some laws and the implementation of programs [76]. However, the programs were designed to intervene with at-risk women, and have limited coverage with no setting appropriate indicators. In some countries with a higher level of medical services and higher health literacy, the choice to receive WHSs is entirely personal, and the governments have not placed a higher priority on these services, or in some cases the intervention would cease or be reduced. This was reflected in the limited government health objectives and data related to the services [77]. WHSs were important guarantees to improve women’s health, and in the context of achieving the SDGs’ target. It is necessary to provide suitable services with increased attention, coordination and integration at the government level [78].

Much evidence has shown that sufficient WHSs, especially maternal healthcare services, could significantly reduce MMR [79,80,81]. In this study, due to the improvement of women’s healthcare, MMR in SH showed a declining trend and was on par with the level of top developed countries [82]. SH was able to achieve this by promoting equal access, to healthcare service packages and by developing detailed service criteria and guidelines, for both urban and rural pregnant women. Specifically, the first measure involved setting priorities for prevention services. A Five-Color Classification Management was implemented to conduct early pregnancy risk assessment and monitor maternal health issues, which meant entering different risk classification and receiving different services [68]. The second measure was to establish a citywide network for critically ill pregnant women. Five municipal-level emergency maternal rescue centers were responsible for interfacing with the 16 districts under their jurisdiction to ensure that the network could respond in a timely manner and effectively treat all critically ill pregnant women [83]. The third measure involved all the hospitals with obstetrical department in SH setting up an Obstetric Safety Office to ensure the standardized obstetric check-ups and safe delivery for pregnant women, and to reduce barriers to consultations and referrals for critically ill pregnant women [84]. With these practices, SH has become a model for WHSs provision in China, and the former National Health and Family Planning Commission has promoted these measures nationwide. The focus of the follow-up work is to further refine and improve the practices. Other countries and regions may refer to such practices in SH to strengthen maternal risk screening.

Compared with SH, the downward trend of MMR in NYC was less obvious and fluctuated. This may be related to the differences in policy priorities, government accountability, and organization structures between the two cities. Serving women’s health and reducing MMR were considered important responsibilities to ensure people’s wellbeing by governments in SH from the municipal to the district level. The latter was chosen as an indicator in the government performance evaluation system. In addition, SH always paid attention to the supervision of the process and effectiveness of WHSs [40]. For example, the obstetric emergency care grading system and responsibility chain was implemented for the management of critically ill pregnant women. A series of regulations, such as case investigation and review, accountability, and notification mechanisms for maternal deaths were set up. Professionals’ performance evaluation was directly linked maternal deaths, which means the supervision department could hold institutions or personnel accountable for avoidable deaths caused by the violation of operating regulations. In NYC, the Department of Health and Mental Health is responsible for reviewing and supervising service quality in the health sector. It is also mandatory that health sectors disclose data related to maternal deaths, but no strict accountability mechanism has been established for maternal deaths. Under the stricter service quality control and accountability mechanisms, it was found that MMR in SH was lower than that in NYC. Therefore, after the Millennium Development Goals were signed, establishing an effective accountability mechanism to promote maternal mortality reduction has become an important issue of the United Nations [85]. Furthermore, organizational system of WHSs continued to improve in SH. There were specialized agencies such as the Maternal and Child Health Centers to provide various categories of maternal services, as well as tertiary care networks including hospitals with clinical departments, district-level medical expert groups, and municipal emergency care centers to take care of critically ill pregnant women. In NYC, the provision of WHSs was dominated by general practitioners, without an emergency network for critically ill pregnant women [86], resulting in worse efficiency and effectiveness of maternal services provision compared to SH. In addition to the factors mentioned above, the differences between the MMRs of the two cities were also related to the socio-economic situation, population structure, etc. [87,88].

The study also showed that neither SH nor NYC provided healthcare services for common menopausal issues such as menopausal depression and perimenopausal syndrome. Taking menopausal depression as an example, after a woman enters menopause, there is a series of symptoms or manifestations such as low mood and slow thinking due to a drop in estrogen levels. Due to the hidden nature of menopausal depression, it is often easily overlooked by women themselves and their families. Those with menopausal depression often feel pressured and lose interest in life, and even choose to commit suicide in the end if their depression is not attended to in time. Currently, menopausal depression shows a clear upward trend, between 25% and 35% in China [89], which has become a public health issue that must be faced and needs to be controlled. Since clinical treatment is still the main measure to deal with menopausal depression, setting up clinics and conducting psychological counseling to help patients with their symptoms, will help those suffering with menopausal depression [90]. There is still a lack of prevention strategies and interventions. Community intervention should be an important measure to prevent menopausal depression, e.g., incorporating the prevention and control of menopausal depression into basic public health services is an implementation path in SH. Through health promotion and education programs, women and their families could have a deeper understanding of the disease symptoms, mechanisms, harms, and treatments [91]. By increasing the social activities of menopausal women, such as participating in card games, dancing, and other physical exercises, their emotions can be effectively regulated. Through the provision of screening services, there can be regular monitoring of women who are at an earlier status of menopausal depression. This can help in intervention or early treatment reduce the harm caused by the disease [92].

This study evaluated the level of WHSs in SH and NYC based on ESCR and AESCR. However, there were a few limitations in the study. First, this study mainly used public policy documents to evaluate the delivery and evolution of WHSs in the two cities. Although we have established uniform standards for content analysis and used the test-rest reliability method with ICC to control the credibility, there is inevitably bias in human judgment. Meanwhile, the omission of unpublished policy documents could not be ruled out, which might affect the research results. Second, the study only selected MMR as the indicator of WHSs in correlation analysis, because of the limited data available for consecutive years. Indicators such as premarital disease screening rate and maternal management rate can also be selected for follow-up quantitative analysis. Third, the decline in the MMR might be influenced by many other factors. Further studies need to verify other socio-economic or medical factors to clarify their contributions to the disparity of MMRs. Fourth, the study only evaluated the WHSs delivery through content analysis of the documents. Indicators from the users and community members’ perspective, such as service utilization and population coverage, can also be used to give a more complete picture of the WHSs delivery in the future.

## 5. Conclusions

This study provides data regarding the provision of WHSs in SH and NYC over the past 40 years, which can contribute to the development of policies to further optimize the services. During the past 40 years, significant progress has been made in the provision of WHSs in the two cities. The MMR has decreased significantly with government-led and delivery of high-quality WHSs.

The provision of WHSs and the promotion of women’s health still need continuous attention from SH and NYC’s governments. As a typical representative of developing and developed countries, it remains challenging to achieve full coverage of WHSs both in SH and NYC. For example, services such as menopause care is in urgent need to be put on the agenda.

The experience of SH and NYC showed that placing WHSs among policy priorities is effective in improving service status, even under different political systems and policy environments. It facilitates the establishment of service networks and the rational allocation of resources, leading to the active participation of both of service providers and the public. The experience also demonstrated that full coverage of essential WHSs should be achieved to improve women’s health from a complete life-cycle management perspective. Extensions to adolescence and menopause are necessary while focusing on maternal health. Furthermore, the quality of services needs to be emphasized, and the population coverage rate and technical specifications should be evaluated to improve WHSs delivery.

Other countries and regions, especially developing countries, can learn from the efforts made in SH and NYC regarding to WHSs and do more to provide a wider variety and coverage of WHSs and reduce MMRs to achieve the SDGs’ target.

This study demonstrates the feasibility of analyzing changes in the delivery of WHSs by means of content analysis. In particular, the use of public documents to assess the policy environment is innovative and the exploration of quantitative evaluation for WHSs in this study could be a new method for assessing the performance of the healthcare systems.

## Figures and Tables

**Figure 1 ijerph-18-04261-f001:**
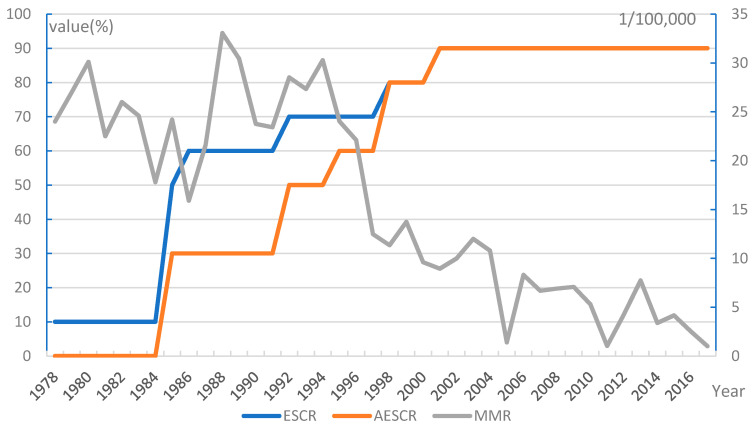
Trends of MMR, ESCR, and AESCR in SH. SH has introduced policies for 9 categories of essential WHSs (i.e., all except menopause care services). All these services have corresponding assessment indicators or assessment criteria. MMR: Maternal mortality ratio; ESCR: The essential maternal healthcare service coverage rate; AESCR: The assessable essential healthcare service coverage rate; SH: Shanghai; WHSs: Women’s healthcare services.

**Figure 2 ijerph-18-04261-f002:**
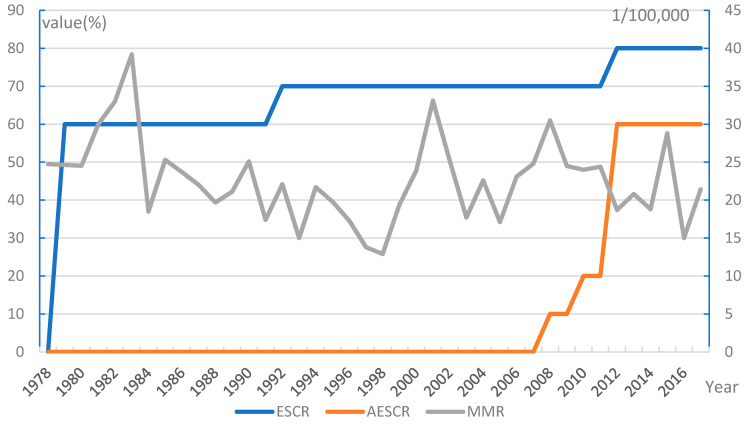
Trends of MMR, ESCR, and AESCR in NYC. NYC has introduced policies on 8 categories of essential WHSs (i.e., except pre-marital healthcare and menopause care services). Infertility intervention and intrapartum care services have no corresponding assessment indicators or assessment criteria in public policy documents. MMR: Maternal mortality ratio; ESCR: The essential maternal healthcare service coverage rate; AESCR: The assessable essential healthcare service coverage rate; NYC: New York City; WHSs: Women’s healthcare services.

**Table 1 ijerph-18-04261-t001:** Definition of 10 Categories of Essential Women’s Healthcare Services (WHSs).

Service Category	Main Contents and Measures
Unintended Pregnancy Prevention	Contraceptive provision, health promotion and health education of safe, effective and appropriate contraceptive methods, etc.
Safe Abortion	Sexual health education, contraceptive provision, legal abortion service provision, etc.
Pre-marital Healthcare	Sexual health education, health promotion and health education of fertility and genetic disease, screening for diseases affecting fertility such as mental illness, mother-to-child transmitted disease and genetic disease.
Family Planning	Consulting services for basic knowledge of pre-pregnancy care, pre-pregnancy medical examination (find out if you have problems affecting fertility/unsuitable for childbearing and related diseases affecting child health), health education (healthy lifestyle for preparing pregnancy, guidance on appropriate timing of conception, etc.).
Infertility Intervention	Infertility consultation, infertility diagnosis and treatment, psychological counseling related to infertility.
High-Risk pregnancy Screening	Screening for pregnancy risk factors, management of high-risk pregnant women (pregnant women with ectopic gestation, oligoamnios, malpresentation, placental abnormality, pregnancy hypertension, gestational diabetes, premature delivery, spontaneous abortion, etc.).
Prenatal Care	Routine prenatal check-ups, prevention of mother-to-child transmission disease (such as AIDS, syphilis, Hepatitis B, etc.), Prenatal screening (such as Down syndrome screening), screening for birth defects, maternal nutritional diseases, etc.
Intrapartum Care	Identifying and managing dystocia, postpartum bleeding management, postpartum hemorrhage prevention, puerperal infections prevention, neonatal asphyxia prevention, prevention of birth canal laceration and neonatal birth injury, childbirth monitoring, etc.
Postpartum Care	Prevention of postpartum complications (such as postpartum hemorrhage, sepsis, eclampsia, etc.), screening and management of postpartum depression, postpartum family visits, health education of breastfeeding, baby nutrition, etc.
Menopause Care	Prevention and intervention of menopausal depression and perimenopausal syndrome.

**Table 2 ijerph-18-04261-t002:** Analysis of the Relationship Essential Women’s Healthcare Service Coverage Rate (ESCR), Assessable Essential Healthcare Service Coverage Rate (AESCR), and Maternal Mortality Ratio (MMR).

Indicators	Correlation Analysis	
	*r*	*p*-Value
ESCR ^1^		
SH ^2^	−0.826	<0.01
NYC ^3^	−0.324	0.041
AESCR ^4^		
SH	−0.835	<0.01
NYC	−0.036	0.826

^1^ ESCR: The essential maternal healthcare service coverage rate; ^2^ SH: Shanghai; ^3^ NYC: New York City; ^4^ AESCR: The assessable essential healthcare service coverage rate.

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
