# Peer review of "Are Essential Women’s Healthcare Services Fully Covered? A Comparative Analysis of Policy Documents in Shanghai and New York City from 1978–2017"

_ijerph, 2021, doi:10.3390/ijerph18084261_

Round 1

Reviewer 1 Report

I found this an interesting study. It is different to the usual studies I am asked to review. The study found that SH recorded better maternal health outcomes compared to NYC, especially dramatically decreased MMR in SH. Yet, the paper might need improvement; methods and interpretations of this study.

[Major revision]

  1. page 3 line 123. I cannot understand the costs in parenthesis in the methods. Presumably, they are in China currency, but it makes the readers confusing. Please present the GDP per capita both cities as USD.
  2. 2. page 3 line 117-133. The authors explained why they choose these two cities but I still wonder curious aspects. The authors presented that these two cities have similar economic and social status but there was no evidence, excepting for GDP (I think it also have no reference in the manuscript). Do SH and NYC have similar structure of population? such as numbers of population, race, or distribution of maternal age. Also, what about healthcare coverage system? Do SH and NYC have similar healthcare coverage (operating on private or public healthcare insurance? or tax?). The authors should explain this information in the study design and setting or add to supplementary files, and prove it these two cities have similar characteristics to appropriate analyzing.
  3. 3. page 4 line 143-159. There was no reference of indicators (DEMHS, AEMHS, ESCR, AESCR). These indicators are important variables in this study. Please explain and clarify these indicators criteria and add up the references. 
  4. 4. page 8 line 280. In methods section, the authors presented that the regression models controlled GDP per capita (page 5 line 200-201) but there was no information in the table 2. Please add up this information in the table 2. Also, did you control only GDP in the linear regression model? I think there was lots of potential confounders such as maternal age, parity, etc. How did the authors adjusted potential confounders? I think current study’s database does not have these information because of the characteristics of administrative document. The authors already wrote it down in the limitation but more information should be explained. 
  5. 5. page 9 line 333-338. I think that this interpretation is a bit unreasonable because the database has lots of limitation. For examples, in this study, we do not know SH and NYC have same characteristics, such as distribution of maternal age, race, parity, healthcare coverage system, etc. However, the authors guessed that the Mayors’ performance assessment as the mechanism. In my opinion, it is unreasonable to interpret to compare these two cities directly without adjusted potential confounders. Also, it cannot be sure, even through NYC doesn't have Mayors' performance assessment, it might be know through a next mayoral election if the Mayor has low performance during the mayor's term. Therefore, I would like to recommend that the author revise these parts more scientific mechanism.
  6. page 9 line 360-384. The authors explained about menopausal issues but there was no result in this study. If the authors discuss this issue, please show the readers the results of this relationship as table or figure.

[Minor revision]

  1. page 1 line 42. 'wom-en's' I think it is a typo.
  2. page 8 line 280. table 2. In 'regression analysis a', 'a' might be a typo. Please add up what you adjusted for confounding variables in this model. What is 'r' mean? is it coeffiecent? I think it is better to present a 'β'.

Author Response

Dear reviewer,

Thank you very much for reviewing our manuscript and your constructive comments.

We have addressed the comments one by one and edited the manuscript accordingly. All of our revisions in the main text are highlighted in red. In the letter, we also provide our point-by-point responses to your comments. 

Please see the attachment for details.

Reviewer 2 Report

Dear Authors,

The paper addresses an important and interesting topic of women’s healthcare services on the example of Shanghai and New York, however its quality and scientific significance is very low. First of all, the authors present in a limited way the background information about both Shanghai and New York women’s healthcare services comparing to the global market and trends. The article lacks the broad literature review and the critical analysis of the main national documents considering the analysed topic. The justification of the study need and its significance is not satisfying. The authors do not present the precise source of obtained data. The obtained results were not discussed broadly, especially in the context of exploring the reasons of differences between trends of MMR, ESCR and AESCR in analysed cities. Moreover, the achieved results were not interpreted in perspective of previous studies and the study implications were not discussed in the broadest context. The article lacks the developed conclusions of the study and possible recommendations considering women’s healthcare particularly in analysed cities. Finally, the linguistic style of article is very colloquial.

Author Response

(The authors gave the same response as above.)

Reviewer 3 Report

The topic is very interesting. Access to health systems and women health care is essential. I have a doubt, I do not know if in the applicable legislation to develop the study it is mandatory to have a favorable opinion from an ethics committee. The study does not reflect that it has an opinion of any ethics committee. Other doubts that arise to me: would it be possible to know how many women do not access the care system? and if this were possible for what reasons?

As it is survey data, the quality of these are limited. In the same way, there may also be data that is not collected. There is a homogeneity for the collection of these data in all places. All of this should be addressed in the discussion. In the same way, this can lead to a series of biases that influence the results. This should also be addressed in the discussion section.

Author Response

(The authors gave the same response as above.)

Round 2

Reviewer 1 Report

Overall, the authors revised the manuscript according to all suggestions. They gave thorough explanations of thorough explanations of their changes and the new version is much improved.

Author Response

Dear reviewer,

Thank you very much for reviewing our manuscript again.  

Should you have any further suggestions, please let us know. Your time and consideration are greatly appreciated.

Reviewer 2 Report

Although some recommendations were corrected by authors, but still there are many remarks that need to be considered by authors.

In my opinion, the authors should expand the research background by presenting the data of women’s healthcare services in Shanghai and New York comparing to the global market and trends during analysed period.

The literature review should be well developed. The authors only mentioned the topic of previous research, but they do not present the main results.

The article should include the broad review of the policies considering the women’s healthcare services using the main national documents in the analysed period as the authors indicate that “The focus of this study is to examine the evolution of women’s health services policies in SH and NYC…” . Therefore, the authors should describe more healthcare system in the analysed period in both cities.

The authors should explain why thy decided to choose Shanghai and New York in the analysis. They claim that Shanghai and New York have similar economic and social status, but considering GDP the cities have totally different economic status. The authors do not explain if the cities have similar structure of population.

Line 333-334 The authors should explain broadly the basis of the following conclusion “Both ESCR and AESCR in NYC are lower than SH. The main reason for this is that NYC did not have a policy requirement for pre-marital healthcare”. What is the pre-marital healthcare? What is the basis of the conclusion that the lower indicators of ESCR and AESCR in NYC are caused by the lack of for pre-marital healthcare? In my opinion it is unreasonable to interpret the results only considering the lack of pre-marital healthcare. The authors should examine deeply the reasons of differences between two cities.

Please, explain the sentence: „New York State did not have health-related regulations for obtaining marriage license”.

The article conclusions possible recommendations are still not satisfying.

Still, the style of article is too colloquial, the text should have more scientific sound and justification e.g.: line 58 „Women’s health is the basis of the prosperity of a country. It has been of great importance worldwide due to its significant role in the sustainable development of society”.

In the text there are many spelling or numeric mistakes, so the authors should read, check and correct the whole article e.g.

  • Line 75-76 „This initiative enabled Sweden to reduce its MMR to 228 per 100,000 live births 75 in the early 20th century, which was the lowest in Europe at that time.” – wrong century
  • Line 81-81 „In Peru, 87% of 81 the pregnant women have access to prenatal care services at least four times, but the coverage rate of prenatal care services was only 12% in Ethiopia” – the lack of the year of presented data.

Author Response

Dear reviewer,

Thank you very much for reviewing our manuscript again.

We have addressed the constructive comments one by one and revised the manuscript accordingly. All of our revisions in the main text are highlighted in red. We also provide our point-by-point responses to your comments.

Please see the attachment for details.
